# A protein microarray analysis of amniotic fluid proteins for the prediction of spontaneous preterm delivery in women with preterm premature rupture of membranes at 23 to 30 weeks of gestation

Hyeon Ji Kim[1], Kyo Hoon Park[1]*, Yu Mi Kim[1], Eunwook Joo[1], Kwanghee Ahn[1], Sue Shin[2]

1 Department of Obstetrics and Gynecology, Seoul National University College of Medicine, Seoul National University Bundang Hospital, Seongnam, Korea, 2 Department of Laboratory Medicine, Seoul National University College of Medicine, Seoul National University Boramae Hospital, Seoul, Korea

* pkh0419@snubh.org

## Abstract

### Objective

We sought to identify novel biomarkers in the amniotic fluid (AF) related to imminent spontaneous preterm delivery (SPTD) ($\leq$ 14 days after sampling) in women with early preterm premature rupture of membranes (PPROM), using a protein microarray.

### Method

This was a retrospective cohort study of a total of 88 singleton pregnant women with PPROM (23+0 to 30+6 weeks) who underwent amniocentesis. A nested case-control study for biomarker discovery was conducted using pooled AF samples from controls (non-imminent delivery, n = 15) and cases (imminent SPTD, n = 15), which were analyzed using an antibody microarray. Quantitative validation of four candidate proteins was performed, using ELISA, in the total cohort (n = 88). IL-8, MMP-9, and Fas levels were additionally measured for the comparison and to examine association of SPTD with the etiologic factors of PPROM.

### Results

Of all the proteins studied in the protein microarray, four showed significant intergroup differences. Analyses of the total cohort by ELISA confirmed the significantly elevated concentrations of AF lipocalin-2, MMP-9, and S100 A8/A9, but not of endostatin and Fas, in women who delivered within 14 days of sampling. For inflammatory proteins showing a significant association, the odds of SPTD within 14 days increased significantly with an increase in baseline AF levels of the proteins (P for trend <0.05 for each) in each quartile, especially in the 3rd and 4th quartile.

**Data Availability Statement:** All relevant data are within the paper and its Supporting Information files.

**Funding:** The current study was supported by the Seoul National University Bundang Hospital Research Fund (Grant No. 13-2020-011), and by the National Research Foundation of Korea (NRF) grant funded by the Korea government (MSIT) (No. 2020R1F1A1048362). The funders had no role in study design, data collection and analysis, decision to publish, or preparation of the manuscript. Hyeon Ji Kim, Kyo Hoon Park, Eunwook Joo, and Kwanghee Ahn have received a salary from Seoul National University Bundang Hospital, because they are employee for Seoul National University Bundang Hospital.

**Competing interests:** The authors have declared that no competing interests exist.

## Conclusions

We identified several potential novel biomarkers (i.e., lipocalin-2, MMP-9, and S100 A8/A9) related to SPTD within 14 days of sampling, all of which are inflammation-related molecules. Furthermore, the SPTD risk increased with increasing quartiles of each of these inflammatory proteins, especially the 3rd and 4th quartile of each protein. The present findings may highlight the importance of inflammatory mechanisms and the degree of activated inflammatory response in developing SPTD in early PPROM.

## Introduction

Preterm premature rupture of membranes (PPROM) remains a significant obstetric problem worldwide that affects nearly 3% of all pregnancies and is responsible for one third of all preterm deliveries [1–3]. PPROM is a major cause of severe neonatal morbidity, mortality, and long-term disability, mainly due to prematurity as well as risks of expectant management for early PPROM (e.g., perinatal infection, umbilical cord accident, and abruptio placentae) [4–6]. Additionally, evidence suggests that gestational age at delivery (or delivery latency) is a more important determinant for short-term neonatal outcome than microbial invasion of amniotic cavity (MIAC) or intra-amniotic inflammation (IAI) in women with PPROM or preterm labor (PTL) [7–9]. Therefore, there is an urgent need to identify predictive biomarkers for the development of spontaneous preterm delivery (SPTD) after PPROM (especially early PPROM $\leq$ 30 weeks).

Over the past decade, several studies have demonstrated that high levels of inflammatory biomarkers in the AF, such as various pro-inflammatory cytokines (interleukin (IL)-6, IL-8) and matrix metalloproteinase (MMP)-8, are associated with SPTD and its pathological mechanisms in women with PPROM [10–14]. However, most studies are limited to identification of only a few specific inflammatory biomarkers and they do not target the total proteins involved in multifaceted complex mechanism for PPROM and subsequent preterm delivery [2]. Furthermore, many of these studies were not designed to evaluate the level of expression based on the concentrations of inflammatory proteins linked to the severity of subsequent SPTD risk, as previously reported in the setting of PTL [9]. In contrast to the studies focusing on a few specific proteins, using multiplex immunoassay, Cobo et al. showed that a higher intra-amniotic inflammatory response, as measured from 24 cytokines, did not predict the occurrence of SPTD within seven days of PPROM [15]. This disagreement between studies may be due to different gestational age groups and a small sample size enrolled within these studies. This highlights the need for a homogeneous group with early PPROM in a large cohort.

Protein-antibody microarray has recently emerged as one of the most promising tools to simultaneously analyze hundreds of proteins in a high-throughput fashion using a limited amount of sample, and has demonstrated promising results for biomarker discovery in the field of complex disease with unknown etiology that may involve multiple proteins. We undertook this study: (i) to assess the protein expression profiles in AF that are associated with imminent SPTD ($\leq$ 14 days) in women with early PPROM, using a protein microarray; (ii) to validate target proteins using quantitative methods in a total cohort of PPROM; and (iii) to examine whether expression levels of these proteins are linked to severity of subsequent SPTD risk.

## Materials and methods

### Study design and participants

This was a retrospective cohort study of women with singleton pregnancies who were admitted to a single tertiary-referral teaching hospital (Seoul National University Bundang Hospital, Seongnamsi, Republic of Korea) with a diagnosis of PPROM from 23+0 to 30+6 weeks gestation, between June 2004 and October 2018. These women underwent amniocentesis to evaluate AF to detect infection or inflammation. The inclusion criteria were as follows: (1) delivery of a live fetus; and (2) availability of an aliquot of AF sample for analysis. Women showing the following characteristics were excluded: (1) medically indicated preterm delivery for maternal or fetal indications within 14 days of sampling; (2) evidence of clinical chorioamnionitis on enrollment; (3) multiple gestations; (4) active labor (defined by the presence of cervical dilatation >3 cm by sterile speculum examination); and (5) major congenital anomalies. The Institutional Review Board Committee approved this study (the Ethics Committee of Seoul National University Bundang Hospital/IRB no. B-1105/128-102), and written informed consent was obtained from all study participants for the amniocentesis procedure, for the collection and use of AF samples, and for the use of their clinical information for research purposes. The primary and secondary outcome measures were SPTD within 14 days and 7 days of sampling, respectively. An additional analysis was performed for preterm delivery at <34+0 weeks.

For the discovery phase using the antibody microarray technique, we performed a nested case-control study on 15 subjects with PPROM who delivered spontaneously within 14 days of sampling (case subjects), and 15 subjects with PPROM who delivered after 14 days (control subjects). These case women were randomly chosen from the list of 45 women who delivered within 14 days of sampling in a cohort of 87 women with PPROM, using a random sequence generator, whose details are described in the S1 File. Each control-woman was chosen for each case-woman, matched by gestational age at sampling, maternal age, parity, and length of specimen storage.

### Sample collection and processing

Transabdominal amniocentesis was performed at the time of admission under sonographic guidance with aseptic conditions. Following previously described methods [16], the AF samples were cultured to identify microorganisms (e.g. genital mycoplasmas (*Mycoplasma hominis* and *Ureaplasma urealyticum*) and aerobic/anaerobic bacteria). Clinicians had access to the AF culture results. The left-over AF was centrifuged for 10 min at 1500×*g* at 4˚C to remove cells and debris, and the supernatant was aliquoted and stored at −80˚C until further use. Medications (e.g., antibiotics, corticosteroids, and tocolytics) were administered after amniocentesis.

### Membrane-based human antibody array

In the discovery phase, we used a Human Antibody Array Kit (AAH-BLM-1B-2; RayBiotech, Norcross, GA, USA) that has the ability to detect 507 different human proteins, including cytokines, chemokines, angiogenic and growth factors, matrix metalloproteases, adipokines, and adhesion molecules, to identify potential target proteins. The methods used for this antibody microarray and image analysis for protein identification have been previously described in detail [17, 18] and are also described in the S1 File. Briefly, AF samples from imminent SPTD case (iSPTD; n = 15, defined as SPTD within 14 days after sampling) and non-imminent SPTD control (non-iSPTD; n = 15) groups were pooled by combining equal amounts (33.3 μg) of 15 individual AF samples from each group. Consequently, 500 μg of AF samples from each

group was mixed and assayed in duplicate, according to the manufacturer's protocol. The signal intensity was detected and quantified by chemiluminescence image analysis (Bio-Rad Quantity 4.6.7; Bio-Rad Laboratories, Inc., Hercules, CA). The signal intensity of each spot was normalized as a percentage of the positive controls on each membrane (after subtraction of local background signals). We used cut-off values of fold change (FC) of $\geq$ 1.5 or $\leq$ 0.66 for up- or downregulated proteins to identify target proteins displaying significant changes in expression by chemiluminescence image analysis.

## ELISA validation

To validate the data obtained from the antibody microarray, we assayed for endostatin, lipocalin-2, the S100 calcium binding protein A8/A9 complex (S100 A8/A9; DuoSet ELISA, R&D Systems, Minneapolis, MN, USA), and the S100 calcium binding protein A100 (S100 A10; Aviva Systems Biology, San Diego, CA, USA) using ELISA kits in the total cohort (n = 88). All ELISAs were carried out as per the manufacturers' instructions. The standard curve range for each of the ELISAs and their dilution ratios are described in detail in the S1 File. The intra- and inter-assay coefficients of variation (CV) were < 10% for the analyzed proteins, except for S100 A8/A9; the inter-assay CV was 14.5%. S100 A10 is not detected in AF, even at dilutions of 1:4, although dilution linearity and spike/recovery testing showed minimal matrix effects and accurate quantitation in the AF samples; therefore, this protein was subsequently excluded from further analysis.

## Additional determination of AF Fas, IL-8, and MMP-9 levels

For comparison with candidate markers, AF IL-8 levels were also measured, using an ELISA human IL-8 DuoSet Kit (R&D System, Minneapolis, MN). Similarly, AF Fas (fibroblast-associated, TNFRSF6) and MMP-9 levels were assayed to test whether the possible mechanisms leading to PPROM are associated with imminent SPTD (Fas for apoptosis and MMP-9 for collagenolysis and extracellular matrix degradation) [2, 19, 20], using an ELISA human DuoSet Kits (R&D System, Minneapolis, MN). The intra- and inter-assay CVs were each below 10%, with the exception of MMP-9 (13.8%).

## Definition, diagnosis, and management of PPROM

PPROM was defined as the leakage of AF occurring prior to 37 weeks before the onset of labor and was visually diagnosed by examination with a sterile speculum to confirm both pooling of AF in the vagina (or leakage of fluid through the cervix) and a positive nitrazine test. Based on previous reports [21, 22], women with PPROM $\geq$ 23+0 weeks and $\leq$ 30+6 weeks of gestation were defined as having early PPROM. Digital examinations were not performed. Management of PPROM, pathological diagnosis of chorioamnionitis, and clinical chorioamnionitis has been previously described in detail [23, 24] and is also described in the S1 File. In brief, prophylactic broad-spectrum antibiotics (ampicillin plus either azithromycin or erythromycin) were administered to all women with PPROM. Antenatal corticosteroids were administered to mature fetal lungs when PPROM occurred between 24+0 and 34+0 weeks of gestation. Tocolytic therapy (magnesium sulfate, ritodrine, or atosiban) was administered in women with PPROM at less than 34 weeks of gestation at the discretion of the attending obstetrician. In most patients with culture-proven MIAC, labor was not induced or an elective cesarean delivery was not performed purely for positive AF cultures. Labor was induced, based on the clinical sign of chorioamnionitis, gestational age, and fetal compromise. Moreover, labor induction was performed after 34 weeks of gestation. MIAC was defined as the presence of a positive AF culture for bacteria, fungi, and/or *Mycoplasma hominis or Ureaplasma* spp. Gestational age

was established based on both first- or second-trimester ultrasound scan and the patient's last menstrual period.

## Statistical analyses

Mann-Whitney $U$-tests or Student's $t$-tests based on the sample distribution test (i.e., Shapiro-Wilk normality test) were used for the comparison of continuous variables, while Fisher's exact tests or $\chi^2$-tests were used for comparison of categorical variables, where appropriate. Thereafter, multivariate logistic regression analyses were performed to examine the independent association between the AF levels of target proteins and SPTD within 14 days and 7 days, while adjusting for baseline clinical parameters (e.g., gestational age at sampling), with a $P$ value < 0.05 from the univariate analysis. Receiver operating characteristic (ROC) curves for predicting SPTD were used to identify optimal cut-off values based on the maximum Youden index [maximum (sensitivity + specificity—1)] for each studied protein in the AF, to assess the diagnostic accuracy of each protein, and to compare the usefulness of different proteins in the same patients using the method proposed by DeLong et al [25]. Finally, we divided the expression level of each protein into quartiles and conducted logistic regression analyses to obtain crude and adjusted odds ratios (AOR) with 95% confidence intervals (CI) to evaluate the association of each quartile with SPTD within 14 days. In brief, for each protein, we compared each quartile to the lowest quartile as the reference category to determine whether increasing levels of AF biomarkers led to an increased risk of SPTD. A Kaplan-Meier survival curve was also constructed for the sampling-to-delivery interval and log-rank tests were used to evaluate differences of delivery-latency between the two groups. Cox regression analyses were used to evaluate the independent relationships between the sampling-to-delivery interval and the covariates after adjusting for other potential confounders. Spearman rank correlation tests were used to measure the relationship between non-normally distributed continuous variables. A two-tailed $P$ value of < 0.05 was considered to be statistically significant. Data were analyzed using SPSS 25.0 (IBM, Armonk, NY, USA).

## Results

### Discovery phase using protein microarray technology

The baseline demographic and clinical characteristics of the discovery cohort are shown in S1 Table. Owing to matching, iSPTD cases and non-iSPTD controls were similar in terms of maternal age, parity, gestational age at sampling, and medication use. However, coincidentally, the proportion of positive AF cultures was significantly higher for the non-iSPTD controls.

Fig 1 shows the results of the protein–antibody microarray experiments in the pooled AF samples from the iSPTD and non-iSPTD groups. When using the criterion ≥ 1.5 or ≤ 0.66 FC in chemiluminescent signal intensity, significant differences were seen between the two groups for only 4 out of 507 human proteins included in the membrane-based microarray (Fig 2). Three proteins were upregulated in AF samples from women with iSPTD compared with those in women without iSPTD; these included lipocalin-2, S100 A8/A9, and S100 A10, with a density ratio between the groups of 1.7, 1.8, and 1.6, respectively (Fig 2). Endostatin expression was downregulated in the iSPTD group compared to that in the non-iSPTD group, with a density ratio of 0.5 between the groups. Although Fas, IL-8, and MMP-9 did not satisfy the selection criterion for the validation study (a 1.1-FC for IL-8, a 1.1-FC for Fas, and a 1.2-FC for MMP-9), their expression levels were additionally assayed, as described in the Methods section.

**PPROM women without iSPTD**

**PPROM women with iSPTD**

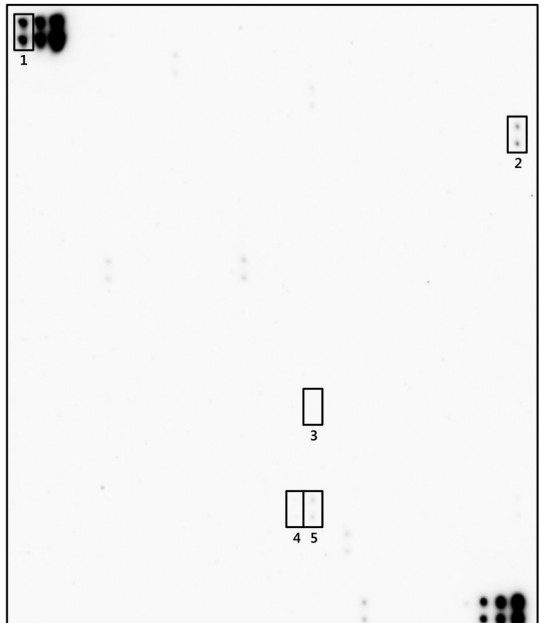
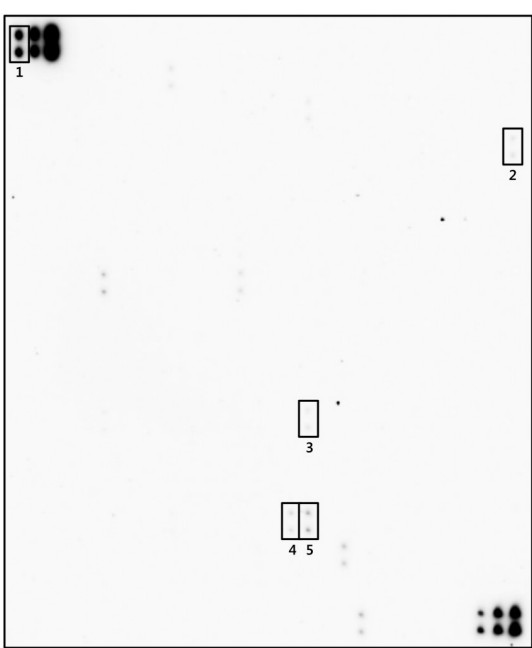

**Fig 1. Expression levels of 507 inflammatory and immunoregulatory proteins in amniotic fluid (AF) from women with preterm premature rupture of membranes (PPROM), comparing the imminent spontaneous preterm delivery (iSPTD) and non-imminent SPTD groups.** The pooled AF samples in each group (15 women with iSPTD, 15 gestational age-matched women without iSPTD) were assayed using a human antibody array kit (AAH-BLM-1B-2; RayBiotech, Norcross, GA). Using cut-off values of fold change of $\geq$ 1.5 or $\leq$ 0.66 for upregulated or downregulated proteins, 4 differentially regulated proteins in AF samples from women with iSPTD relative to women without iSPTD are indicated in rectangles. Number 1 shows the positive controls.

### ELISA validation in the total cohort

In the total cohort, SPTD was 51.1% (45/88) within 14 days of sampling and 39.8% (35/88) within 7 days. Unlike the discovery cohort, where the clinical parameters were similar between the two groups, women delivering within 14 days had a significantly lower mean gestational age at sampling and higher rates of tocolytic therapy and positive AF cultures than those delivering after 14 days (Table 1). The median levels of lipocalin-2, S100 A8/A9, IL-8, and MMP-9 were significantly higher in AF from women delivering within 14 days after sampling (Table 1).

The multivariable analysis revealed that the elevated levels of AF IL-8, lipocalin-2, MMP-9, and S100 A8/A9 were significantly associated with SPTD within 14 days of sampling, when adjusted for gestational age at sampling and tocolytics administration (Table 2). However, the AF endostatin and Fas levels did not differ significantly between the two groups (Table 1). The AUC values of AF IL-8, lipocalin-2, MMP-9, and S100 A8/A9 for the prediction of SPTD within 14 days after sampling were 0.717, 0.725, 0.755, and 0.714, respectively (Table 3 and Fig 3), and were not significantly different from each other (all variables: $P$ = 0.25–0.92). Levels of AF proteins found to be significantly associated with SPTD within 14 days of sampling (IL-8, lipocalin-2, MMP-9, and S100 A8/A9) were significantly correlated with each other (all factors, r = 0.59–0.85, $P$ < 0.001).

Table 4 summarizes the crude and adjusted odds ratios for SPTD within 14 days of sampling in PPROM women across the quartiles of each protein. Overall, the odds of SPTD occurring within 14 days of sampling increased significantly with each quartile increase in baseline

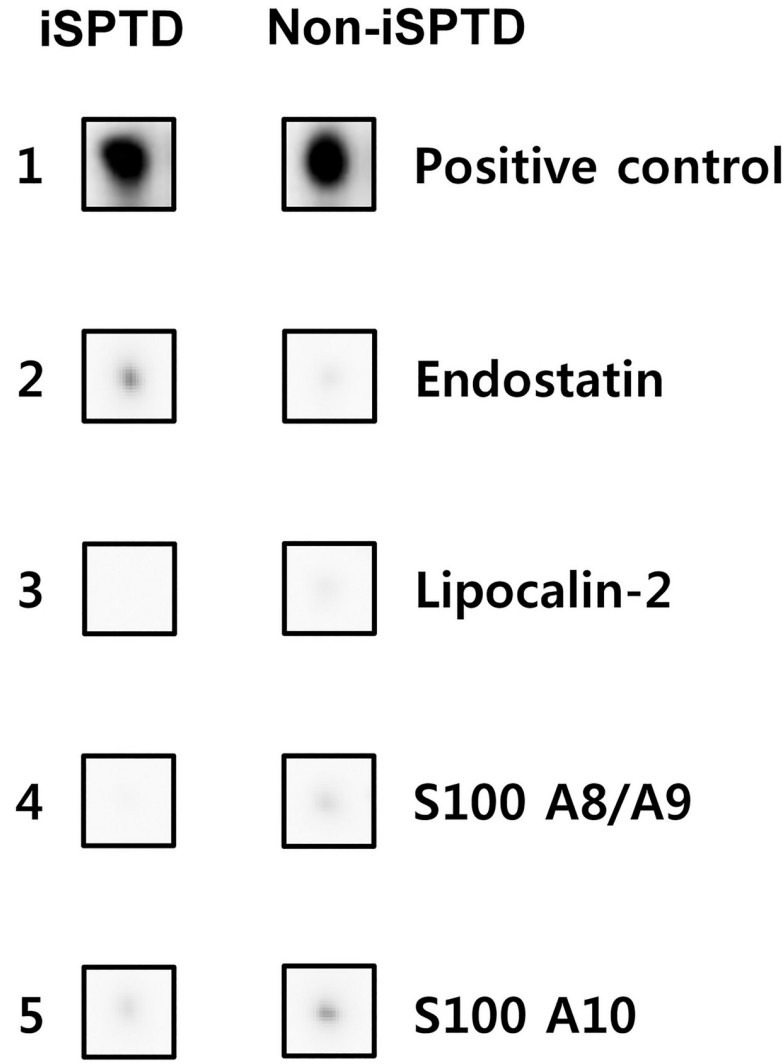

**Fig 2. List of 4 proteins with significantly differential expression between the amniotic fluid (AF) from women with preterm premature rupture of membranes (PPROM), who had subsequent spontaneous preterm delivery (SPTD) within 14 days of sampling and AF from PPROM women delivering after 14 days, using the criteria of a $\geq$ 1.5-fold change or a $\leq$ 0.66-fold change in signal intensity.** iSPTD, imminent spontaneous preterm delivery.

AF levels of each protein ($P$ for trend < 0.01 for each), especially with the 3rd and 4th quartile of each protein. For example, PPROM patients in the highest quartile of AF lipocalin-2 were at approximately 12 times higher odds of undergoing SPTD within 14 days of sampling compared to those in the lowest quartile. This relationship did not change when adjusting for baseline differences in gestational age at sampling and use of tocolytics (Table 4). Similarly, the adjusted OR was 16.9 (95% CI, 3.4–82.6) for the highest quartile of AF lipocalin-2 relative to the lowest quartile, after adjustment for potential confounders.

## New protein biomarkers in AF and the sampling-to-delivery interval

Kaplan-Meier survival analyses revealed that PPROM women with higher AF levels of lipocalin-2 ($\geq$ 0.36 μg/mL; log-rank test, $P$ < 0.001), MMP-9 ($\geq$ 4.27 ng/mL; log-rank test, $P$ < 0.001), or S100 A8/A9 ($\leq$ 10.99 μg/mL; log-rank test, $P$ = 0.004) exhibited significantly

**Table 1. Characteristics of the study population grouped by spontaneous preterm delivery within 14 days of sampling in the total cohort.**

| Variables | Spontaneous preterm delivery after sampling | | P-value |
|---|---|---|---|
| | ≤ 14 days (n = 45) | > 14 days (n = 43) | |
| Maternal age (years) | 31.9 ± 3.7 | 32.1 ± 3.5 | 0.792[a] |
| Nulliparity | 40.0% (18/45) | 46.5% (20/43) | 0.538[c] |
| Gestational age at sampling (weeks) | 28.2 ± 1.9 | 27.1 ± 2.4 | 0.028[b] |
| Gestational age at delivery (weeks) | 28.8 ± 1.8 | 32.6 ± 3.2 | <0.001[b] |
| Sampling-to-delivery interval (days) | 4.2 ± 3.7 | 37.7 ± 21.1 | <0.001[b] |
| AF endostatin (ng/mL) | 69.6 ± 27.4 | 65.9 ± 21.8 | 0.496[b] |
| AF Fas (ng/mL) | 5.17 ± 1.98 | 4.63 ± 1.82 | 0.200[b] |
| AF IL-8 (ng/mL) | 8.3 ± 6.4 | 4.0 ± 5.6 | 0.001[b] |
| AF lipocalin-2 (μg/mL) | 1.51 ± 0.99 | 0.76 ± 0.80 | <0.001[b] |
| AF MMP-9 (ng/mL) | 107.11 ± 94.39 | 30.78 ± 61.09 | <0.001[b] |
| AF S100 A8/A9 (μg/mL) | 26.7 ± 3.5 | 12.3 ± 2.8 | 0.002[b] |
| Positive AF cultures | 60.0% (27/45) | 30.2% (13/43) | 0.005[c] |
| Use of tocolytic agents | 84.4% (38/45) | 55.8% (24/43) | 0.003[c] |
| Use of antibiotics | 95.6% (43/45) | 95.3% (41/43) | 1.000[c] |
| Use of antenatal corticosteroids | 95.6% (43/45) | 83.7% (36/43) | 0.086[c] |
| Clinical chorioamnionitis | 15.6% (7/45) | 14.0% (6/43) | 0.832[c] |
| Histological chorioamnionitis[d] | 68.9% (31/45) | 62.5% (25/40) | 0.535[c] |

AF, amniotic fluid; Fas (TNFRSF6), fibroblast-associated (tumor necrosis factor receptor superfamily member 6); IL, interleukin; MMP, matrix metalloproteinase; S100 A8/A9, S100 calcium binding protein A8/A9 complex.

Data are given as the mean ± standard deviation or % (n/N).

[a] Student's t-tests.

[b] Mann-Whitney U-tests.

[c] $\chi^2$-tests or Fisher's exact tests, where appropriate.

[d] Three cases were excluded for the analysis because delivery took place at another institution.

shorter sampling-to-delivery intervals (Fig 4). Similarly, using Cox proportional hazards regression analyses, high AF levels of lipocalin-2 (hazard ratio: 3.45; 95% CI, 1.99–5.95, $P < 0.001$), MMP-9 (hazard ratio: 3.47; 95% CI, 1.99–6.02, $P < 0.001$), and S100 A8/A9 (hazard ratio: 1.78; 95% CI, 1.12–2.84, $P = 0.015$) were found to be significantly associated with shorter sampling-to-delivery intervals, after adjusting for gestational age at sampling and use of tocolytics.

**Table 2. Multivariable logistic regression model showing the unadjusted and adjusted odds ratios of association between potential amniotic fluid proteins and spontaneous preterm delivery within 14 days in women with preterm premature rupture of membranes in the total cohort (n = 88).**

| Variables | Odds ratio (95% confidence interval) | | |
|---|---|---|---|
| | Unadjusted | Adjusted[a] | P-value[b] |
| AF IL-8 (ng/mL) | 1.126 (1.044–1.215) | 1.134 (1.043–1.232) | 0.003 |
| AF lipocalin-2 (μg/mL) | 2.401 (1.465–3.937) | 2.969 (1.632–5.401) | <0.001 |
| AF MMP-9 (ng/mL) | 1.011 (1.005–1.017) | 1.012 (1.005–1.019) | <0.001 |
| AF S100 A8/A9 (μg/mL) | 1.033 (1.011–1.056) | 1.034 (1.010–1.059) | 0.006 |

AF, amniotic fluid; IL, interleukin; MMP, matrix metalloproteinase; S100 A8/A9, S100 calcium binding protein A8/A9 complex.

[a] For gestational age at sampling and use of tocolytics.

[b] Of odds ratio adjusted for gestational age at sampling and use of tocolytics.

**Table 3. Diagnostic indices of lipocalin-2, MMP-9, S100 A8/A9, and interleukin-8 in amniotic fluid to predict spontaneous preterm delivery within 14 days of sampling in women with preterm premature rupture of membranes in the total cohort (n = 88).**

| Variables | Area (± SE) under the ROC curve[a] | 95% CI | Cut-off value[b] | Sensitivity[c](95% CI) | Specificity[c](95% CI) | PPV | NPV |
|---|---|---|---|---|---|---|---|
| AF IL-8 (ng/mL) | 0.717 ± 0.055 | 0.610–0.825 | ≥ 4.39 | 74.3 (57.8–86.9) | 67.3 (52.4–80.0) | 64.4 | 76.7 |
| AF lipocalin-2 (µg/mL) | 0.725 ± 0.054 | 0.619–0.832 | ≥ 0.36 | 88.4 (70.5–93.5) | 55.8 (39.8–70.9) | 66.6 | 77.4 |
| AF MMP-9 (ng/mL) | 0.755 ± 0.053 | 0.651–0.859 | ≥ 4.27 | 83.7 (69.3–93.2) | 60.5 (44.4–75.0) | 67.9 | 78.8 |
| AF S100 A8/A9 (µg/mL) | 0.714 ± 0.055 | 0.606–0.822 | ≥ 10.99 | 60.0 (44.3–74.3) | 76.7 (61.3–88.2) | 72.9 | 64.7 |

SE, standard error; ROC, receiver operating characteristics; CI, confidence interval; PPV, positive predictive value; NPV, negative predictive value; AF, amniotic fluid; IL, interleukin; MMP, matrix metalloproteinase; S100 A8/A9, S100 calcium binding protein A8/A9 complex.

[a] Receiver operating characteristic curve analysis.

[b] Cut-off values corresponding to the highest sum of sensitivity and specificity.

[c] Values are presented as % (95% CI).

### Validation of the cohort using SPTD within 7 days of sampling as the outcome parameter

When using SPTD within 7 days of sampling as the outcome measure, both univariate and multivariate analyses yielded the same results as those using SPTD within 14 days of sampling (S2 and S3 Tables). The AUC values of AF IL-8, lipocalin-2, MMP-9, and S100 A8/A9 for the prediction of SPTD within 7 days after sampling were 0.737, 0.717, 0.716, and 0.689, respectively (S4 Table and S1 Fig) and were not significantly different from each other (all variables: $P$ = 0.28–0.73).

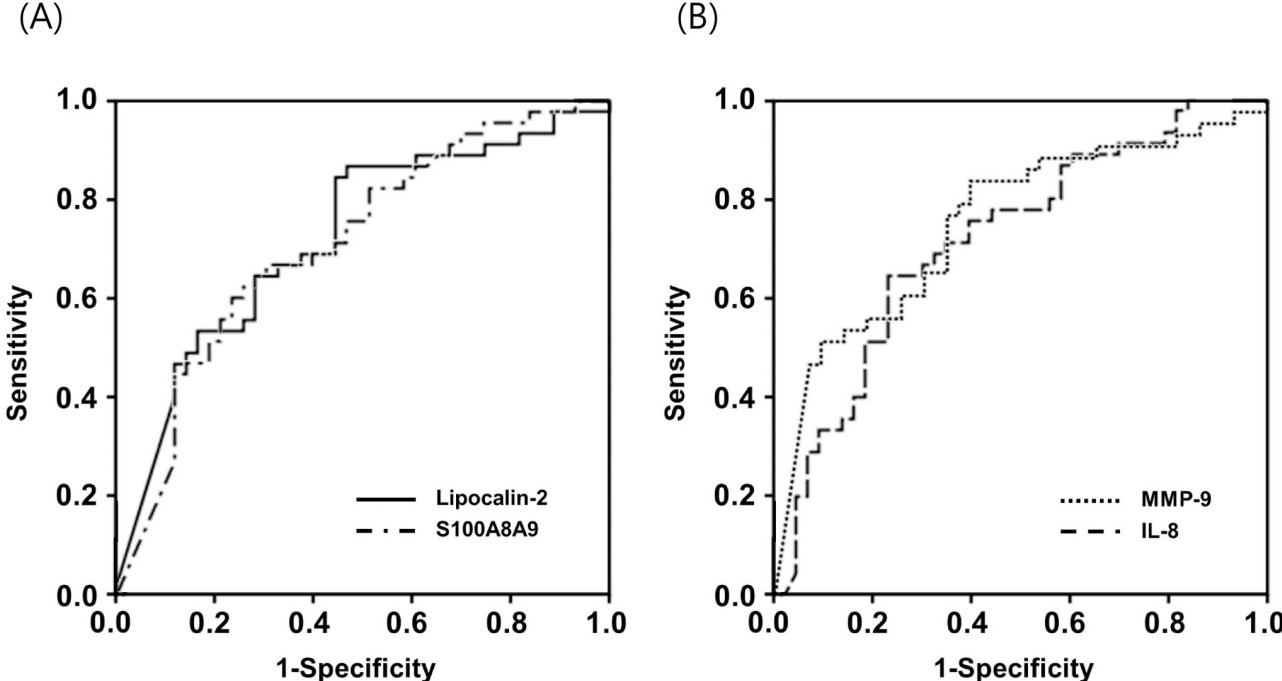

**Fig 3.** (A) Receiver operating characteristic (ROC) curves of amniotic fluid (AF) lipocalin-2 and S100 A8/A9 at predicting spontaneous preterm delivery (SPTD) within 14 days (AF lipocalin-2: area under the curve [AUC] = 0.725, SE = 0.054; and AF S100 A8/A9: AUC = 0.714, SE = 0.055). (B) ROC curves of AF matrix metalloproteinase-9 (MMP-9) and interleukin-8 (IL-8) at predicting SPTD within 14 days (AF MMP-9: AUC = 0.755, SE = 0.053; and AF IL-8: AUC = 0.717, SE = 0.055). Differences among the AUCs of AF lipocalin-2, S100 A8/A9, MMP-9, and IL-8 were not significant (all variables: $P$ = 0.25–0.92). S100 A8/A9, S100 calcium binding protein A8/A9 complex.

**Table 4. The unadjusted and adjusted odds ratios of spontaneous preterm delivery within 14 days of sampling according to quartiles of potential amniotic fluid protein expression in women with preterm premature rupture of membranes in the total cohort (n = 88).**

| | Quartile of IL-8 (range, ng/mL) | | | | |
|---|---|---|---|---|---|
| | 1 (< 0.69) | 2 (0.69–3.50) | 3 (3.50–12.31) | 4 (> 12.31) | *P* for trend |
| Crude analysis | | | | | |
| OR (95% CI) | 1.0 | 2.83 (0.77–10.43) | 5.95 (1.59–22.33) | 9.07 (2.31–35.65) | 0.001[a] |
| | · · · | 0.117 | 0.008 | 0.002 | |
| Gestational age at sampling and use of tocolytics adjusted | | | | | |
| OR (95% CI) | 1.0 | 2.40 (0.59–9.75) | 9.38 (2.08–42.24) | 8.48 (1.88–38.20) | 0.001[a] |
| *P* | · · · | 0.220 | 0.004 | 0.005 | |
| | Quartile of lipocalin-2 (range, ng/mL) | | | | |
| | 1 (< 261.45) | 2 (261.45–710.43) | 3 (710.43–2500.00) | 4 (>2500.00) | *P* for trend |
| unadjusted analysis | | | | | |
| OR (95% CI) | 1.0 | 2.83 (0.77–10.43) | 4.53 (1.21–16.96) | 12.24 (3.00–49.92) | <0.001[a] |
| *P* | · · · | 0.117 | 0.025 | <0.001 | |
| Gestational age at sampling and use of tocolytics adjusted | | | | | |
| OR (95% CI) | 1.0 | 2.65 (0.66–10.66) | 5.14 (1.20–21.99) | 16.99 (3.49–82.68) | <0.001[a] |
| *P* | · · · | 0.171 | 0.027 | <0.001 | |
| | Quartile of MMP-9 (range, ng/mL) | | | | |
| | 1 (< 1.36) | 2 (1.36–12.97) | 3 (12.97–200.0) | 4 (> 200.0) | *P* for trend |
| Crude analysis | | | | | |
| RR (95% CI) | 1.0 | 2.40 (0.64–9.03) | 2.40 (0.64–9.03) | 21.33 (4.42–103.07) | <0.001[a] |
| *P* | · · · | 0.195 | 0.195 | <0.001 | |
| Gestational age at sampling and use of tocolytics adjusted | | | | | |
| RR (95% CI) | 1.0 | 1.89 (0.45–7.99) | 2.65 (0.61–11.45) | 20.92 (3.84–113.87) | <0.001[a] |
| *P* | · · · | 0.387 | 0.192 | <0.001 | |
| | Quartile of S100 A8/A9 (range, μg/mL) | | | | |
| | 1 (< 3.01) | 2 (3.01–8.16) | 3 (8.16–30.07) | 4 (> 30.07) | *P* for trend |
| Crude analysis | | | | | |
| OR (95% CI) | 1.0 | 1.85 (0.52–6.55) | 3.85 (1.09–13.66) | 9.07 (2.31–35.65) | 0.001[a] |
| *P* | · · · | 0.343 | 0.037 | 0.002 | |
| Gestational age at sampling and use of tocolytics adjusted | | | | | |
| OR (95% CI) | 1.0 | 1.56 (0.39–6.18) | 3.27 (0.85–12.62) | 8.86 (2.00–39.23) | 0.002[a] |
| *P* | · · · | 0.528 | 0.086 | 0.004 | |

OR, odds ratio; CI, confidence interval; IL, interleukin; MMP, matrix metalloproteinase; S100 A8/A9, S100 calcium binding protein A8/A9 complex.

[a] Logistic regression analysis

## AF biomarkers with respect to preterm delivery at < 34+0 weeks

When preterm delivery at < 34 weeks of gestation was used as the outcome measure, univariate analyses yielded the same results for the measured AF proteins as those obtained using SPTD within 14 days of sampling (S5 Table). However, the rates of use of tocolytics and corticosteroids were significantly higher in women who subsequently had preterm delivery at <34 weeks than in women who delivered at ≥34 weeks. Thus, multivariate analyses were performed, which revealed that significant associations between the four proteins in AF and preterm delivery at < 34 weeks, which was corroborated by univariate analyses, disappeared upon adjusting for baseline variables (i.e., use of tocolytics and corticosteroids) in multivariate analyses (S6 Table).

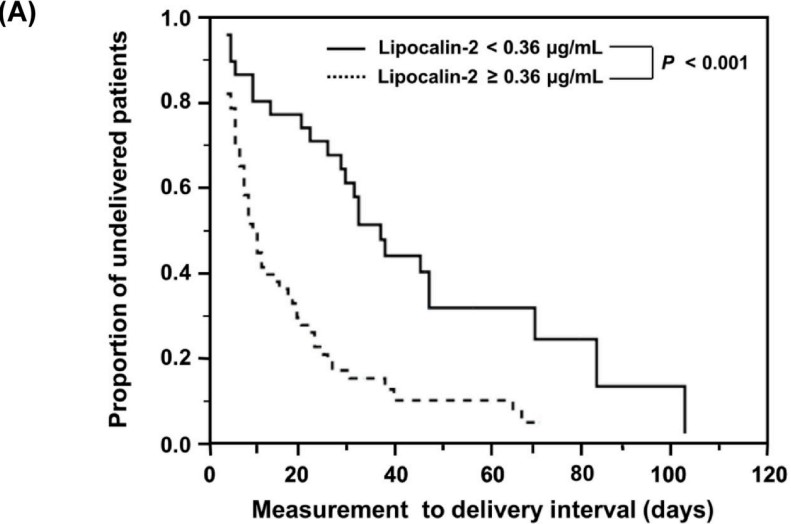

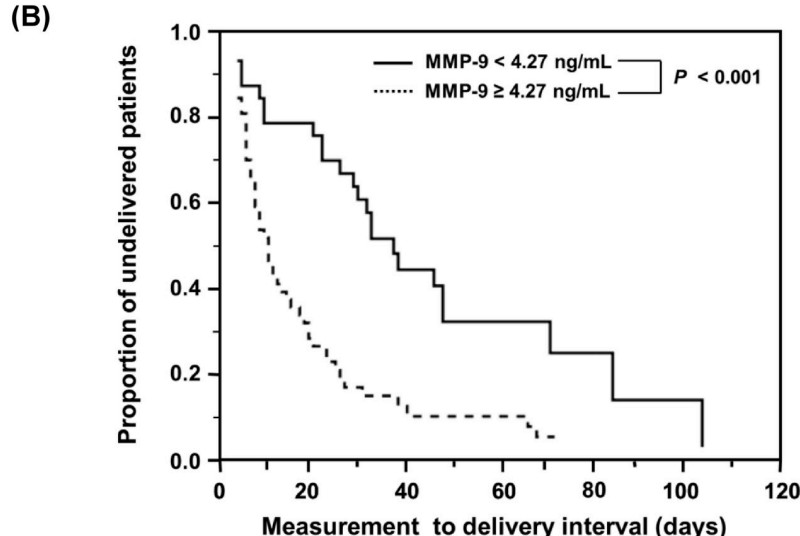

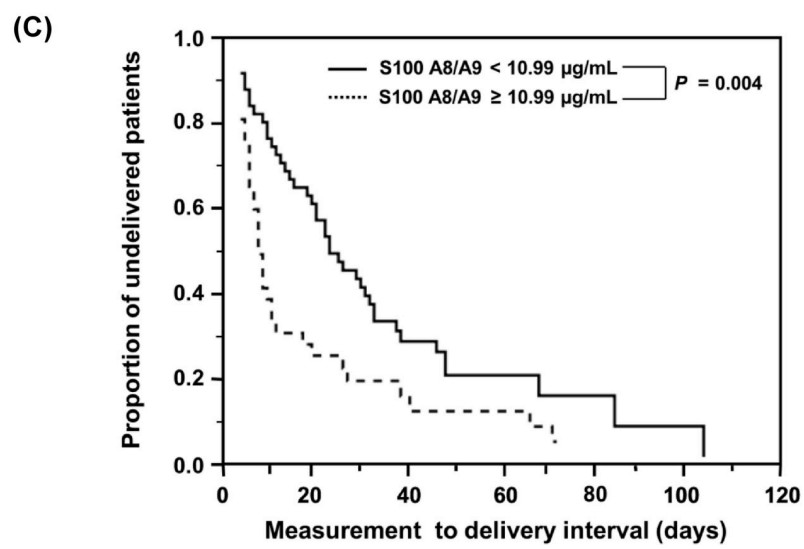

**Fig 4.** Kaplan-Meier survival estimates of the sampling-to-delivery interval for (A) AF lipocalin-2 of $\geq$0.36 or <0.36 µg/mL (median, 7.00 days [95% CI, 4.23–9.78] vs. 36.00 days [95% CI, 27.23–44.78]; $P$ < 0.001), (B) AF MMP-9 of $\geq$ 4.27 or < 4.27 ng/mL (median, 8.00 days [95% CI, 4.84–11.16] vs. 36.00 days [95% CI, 27.21–44.79]; $P$ < 0.001), and (C) AF S100 A8/A9 of $\geq$10.99 or <10.99 µg/mL (median, 5.00 days [95% CI, 3.29–6.70] vs. 21.00 days [95% CI, 14.16–27.84]; $P$ = 0.004). AF, amniotic fluid; CI, confidence interval; MMP-9, matrix metalloproteinase-9; S100 A8/A9, S100 calcium binding protein A8/A9 complex.

## Discussion

### Main findings

In the present study, we identified novel independent AF biomarkers (lipocalin-2, MMP-9, and S100 A8/A9) associated with imminent SPTD in women with early PPROM using protein–antibody microarray analyses and ELISAs. All of these novel biomarkers are inflammation-based and their predictive performances for imminent SPTD were similar to that of AF IL-8. Furthermore, stratified analysis based on the quartiles of each protein expression level revealed that the risks of imminent SPTD increased with increasing quartiles of each of these inflammation-based biomarkers, especially the 3rd and 4th quartile of each biomarker. These data may provide novel insights into the molecular mechanisms by which SPTD occurs after early PPROM and offer a perspective for new strategies of therapeutic intervention to prolong gestational latency.

Evidence suggests that inflammation is an essential component in the initiation of term as well as preterm parturition [26]. Specifically, elevated IL-6, IL-8, and MMP-8 levels in the amniotic cavity have been demonstrated to be associated with the onset of preterm and term labor, along with membrane rupture [16, 26–29]. Furthermore, the present study has shown that the severities of SPTD are associated with the gradations in inflammatory response in the AF based on the levels of multiple inflammatory markers. These findings were in agreement with data from previous studies on women with PTL or PPROM, which have shown that IAI is not simply present or absent but also has a degree of severity, based on AF IL-6 levels (or AF Mass Restricted (MR) scores), that correlated with shorter latencies [9, 30]. Our results have important clinical implications for therapeutic perspectives because women with early PPROM may benefit from treatment to delay or prevent progression to the higher quartiles of inflammation-related proteins in relation to prolonged latency following early PPROM. In fact, clinical trial data have revealed that empirical antibiotic treatment following PPROM significantly prolongs latency until delivery and reduces neonatal morbidity [31]. Further studies are needed to study whether treatment with anti-inflammatory drugs, immune modulators, or corticosteroids used to reduce the pathological effects of the inflammatory response may lead to prolonged gestational latency in women with early PPROM.

The mild degree of inflammation-related protein expression observed in our study may be encountered in women with early PPROM in the following clinical circumstances: (i) low levels of inflammation-related proteins in the AF represent an early stage that progresses to high or increasing levels of these inflammatory proteins and (ii) low AF levels of inflammatory proteins may be maintained and slowly progress over time until delivery. Our data cannot distinguish these two possibilities and further studies based on serial amniocentesis are required to resolve these issues. Of course, we believe that these two clinical situations may depend on the occurrence of a secondary infection in the amniotic cavity as well as the effect of antibiotics against isolated bacteria administered during expectant management. Recently, Lee et al. conducted studies based on serial amniocentesis and proposed a new antibiotic regimen (ceftriaxone, clarithromycin, and metronidazole) which was more effective at treating and eradicating MIAC/IAI, preventing secondary MIAC/IAI, and prolonging the latency period in PPROM [32, 33].

Lipocalin-2 (also known as neutrophil gelatinase-associated lipocalin or siderocalin) is expressed in human neutrophils in response to infection/inflammation and ischemia and is involved in innate immunity and apoptosis [34]. In particular, this protein is produced constitutively in fetal membranes and villous trophoblast and augments collagenolytic effects of MMP-9 through its ability to complex with MMP-9 [35, 36]. Recent studies on AF have revealed that lipocalin-2 is abundantly present in normal AF and its expression is upregulated in AF with MIAC, IAI, or histologic chorioamnioitis [35, 37]. These results are, in general, in agreement with the results of the present study where associations between AF lipocalin-2 and SPTD were found in PPROM, considering reports demonstrating that the occurrence of SPTD in PPROM is also associated with major markers suggesting the existence of intra-uterine infection [38–41].

MMP-9, a member of the gelatinase family of MMPs, was shown to mediate leukocyte migration during inflammation, activate cytokines and chemokines, and directly degrade extracellular matrix proteins [20, 42, 43]. Thus, MMP-9 has been intensively investigated for its role in membrane rupture and its association with *in utero* infection/inflammation. MMP-9 has been shown to mediate the key events leading to membrane rupture in fetal membranes, and MMP-9 expression was also found to be elevated in the AF of PPROM women with MIAC; moreover, to our knowledge, previous research did not investigate the changes in MMP-9 levels in AF associated with imminent SPTD in the context of PPROM. We found that upregulation of AF MMP-9 was independently associated with SPTD within 14 days and 7 days of sampling in women with PPROM.

S100 A8/A9 (also called calprotectin or myeloid-related protein 8/14) is a heterodimeric protein that is constitutively expressed in immune cells such as monocytes and neutrophils [44]. S100 A8/A9 stimulates leukocyte recruitment and induces cytokine secretion during inflammation and is, therefore, recognized as an important modulator of the inflammatory process [44]. Consistent with the known biological functions of S100 A8/A9, increased S100 A8/A9 expression was observed in the AF or plasma from patients with histological chorioamnionitis, IAI, and MIAC in women diagnosed with PTL or PPROM [18, 45, 46]. However, to date, no one has investigated the association between changes in S100 A8/A9 expression in AF and SPTD in women with early PPROM. We have demonstrated that elevated S100 A8/A9 levels in AF are independently associated with SPTD within 14 days and 7 days of sampling.

A body of evidence suggested that fetal membrane apoptosis is one of the important pathologic mechanisms linked to the occurrence of PPROM [2, 20]. Fas, also known as Apo-1 and TNFRSF6, is a cell-surface receptor involved in apoptotic signaling in several cell types; Fas expression was also observed in the prematurely ruptured membranes but not in the preterm labor membranes [20, 47]. Unlike the inflammation-based biomarkers that reflect mechanisms related to the occurrence of PPROM, the present study did not show that upregulated AF Fas was associated with imminent SPTD, despite a report that this protein is elevated in the AF of patients with MIAC [48]. These observations are unexpected given the reported association of MIAC with subsequent SPTD risk [38, 39] and suggest that Fas is not involved in the pathological mechanisms of subsequent SPTD in the context of PPROM.

## Strengths and limitations of the study

The limitations of the current study include its retrospective nature, single-center setting, and relatively small sample size. In addition, owing to the small sample size, the association between 5 potential candidate markers and the primary endpoint was not validated in independent replication cohorts, but it was in the total cohort, although these observed associations are the same as those in the cohort after excluding the patients analyzed in the discovery phase

(S7–S10 Tables). As such, further prospectively designed studies are needed to validate our findings in other cohorts. Another limitation of the study is that the rate of positive AF cultures in the exploratory phase, by chance, was significantly higher in the non-iSPTD controls than in the iSPTD cases, which may have led us to overlook many other inflammatory proteins (e.g., IL-6 and IL-8) as candidate markers, and thus a small number of markers of interest were identified in the current study. Finally, measurements of AF marker levels are limited in clinical practice as they require invasive AF sampling and can be technically difficult in cases with severe oligohydramnios secondary to PPROM. The strength of the current study is that it is the first profiling study to examine protein expression changes in the AF associated with imminent SPTD in women with early PPROM, using a high-throughput protein microarray. Another strength of the present study is that the study participants were homogeneous in treatment based on gestational age at the time of rupture, as we included only women with gestational ages from 23 to 30 weeks. In the current study, we chose SPTD within 14 days of sampling as the primary endpoints. In the context of PPROM, prolonged pregnancy until at least 32 weeks of gestation is preferred especially when the fetus is 23–30 weeks, even though subclinical chorioamnionitis may be present [49].

## Conclusions

In summary, using protein–antibody microarray analyses and ELISAs of AF from women with early PPROM, we have discovered several potential novel biomarkers (i.e., lipocalin-2, MMP-9, and S100 A8/A9) related to SPTD within 14 days of sampling, all of which are inflammation-related molecules. Furthermore, this SPTD risk increased with increasing quartiles of each of these inflammatory proteins, especially the 3rd and 4th quartile of each protein. The present findings highlight the importance of inflammatory mechanisms and the degree of activated inflammatory response in developing SPTD in early PPROM.

## Supporting information

**S1 Fig.** (A) Receiver operating characteristic (ROC) curves of amniotic fluid (AF) lipocalin-2 and S100 A8/A9 at predicting spontaneous preterm delivery (SPTD) within 7 days (AF lipocalin-2: area under the curve [AUC] = 0.717, SE = 0.057; and AF S100 A8/A9: AUC = 0.689, SE = 0.059). (B) ROC curves of AF matrix metalloproteinase-9 (MMP-9) and interleukin-8 (IL-8) at predicting SPTD within 7 days (AF MMP-9: AUC = 0.755, SE = 0.053; and AF IL-8: AUC = 0.737, SE = 0.054). Differences among the AUCs of AF lipocalin-2, S100 A8/A9, MMP-9, and IL-8 were not significant (all variables: *P* = 0.28–0.73). S100 A8/A9, S100 calcium binding protein A8/A9 complex.
(TIF)

**S1 Table. Demographic and clinical characteristics of women with preterm premature rupture of membranes recruited for protein microarray analysis.**
(DOCX)

**S2 Table. Characteristics of the study population grouped by spontaneous preterm delivery within 7 days of sampling in the total cohort.**
(DOCX)

**S3 Table. Multivariable logistic regression model.** Unadjusted and adjusted odds ratios of association between potential amniotic fluid proteins and spontaneous preterm delivery within 7 days in women with preterm premature rupture of membranes in the total cohort.
(DOCX)

**S4 Table. Diagnostic indices of lipocalin-2, MMP-9, S100 A8/A9, and interleukin-8 in amniotic fluid to predict spontaneous preterm delivery within 7 days of sampling in women with preterm premature rupture of membranes in the total cohort.**
(DOCX)

**S5 Table. Characteristics of the study population grouped by preterm delivery at $< 34$ weeks in the total cohort.**
(DOCX)

**S6 Table. Multivariable logistic regression model.** Unadjusted and adjusted odds ratios of association between potential amniotic fluid proteins and preterm delivery at $< 34$ weeks in women with preterm premature rupture of membranes in the total cohort.
(DOCX)

**S7 Table. Characteristics of the study population grouped by spontaneous preterm delivery within 14 days of sampling in the cohort after excluding the patients analyzed in the discovery phase.**
(DOCX)

**S8 Table. Characteristics of the study population grouped by spontaneous preterm delivery within 7 days of sampling in the cohort after excluding the patients analyzed in the discovery phase.**
(DOCX)

**S9 Table. Multivariable logistic regression model.** Unadjusted and adjusted odds ratios of association between potential amniotic fluid proteins and spontaneous preterm delivery within 14 days in women with preterm premature rupture of membranes in the cohort after excluding the patients analyzed in the discovery phase.
(DOCX)

**S10 Table. Multivariable logistic regression model.** Unadjusted and adjusted odds ratios of association between potential amniotic fluid proteins and spontaneous preterm delivery within 7 days in women with preterm premature rupture of membranes in the cohort after excluding the patients analyzed in the discovery phase.
(DOCX)

**S1 Data. Raw data for the total cohort.**
(SAV)

**S2 Data. Raw data for the cohort after excluding the patients analyzed in the discovery phase.**
(SAV)

**S1 File.**
(DOCX)

## Acknowledgments

We are grateful to the patients in the study. We thank our medical staffs for their assistance.

## Author Contributions

**Conceptualization:** Hyeon Ji Kim, Kyo Hoon Park, Sue Shin.

**Data curation:** Hyeon Ji Kim, Kyo Hoon Park, Eunwook Joo, Kwanghee Ahn.

**Formal analysis:** Hyeon Ji Kim, Kyo Hoon Park, Yu Mi Kim, Eunwook Joo, Kwanghee Ahn.

**Funding acquisition:** Kyo Hoon Park.

**Investigation:** Hyeon Ji Kim, Kyo Hoon Park, Yu Mi Kim, Eunwook Joo, Kwanghee Ahn, Sue Shin.

**Methodology:** Hyeon Ji Kim, Kyo Hoon Park, Yu Mi Kim, Eunwook Joo, Kwanghee Ahn.

**Project administration:** Kyo Hoon Park.

**Supervision:** Kyo Hoon Park, Sue Shin.

**Validation:** Kyo Hoon Park.

**Writing – original draft:** Hyeon Ji Kim, Kyo Hoon Park.

**Writing – review & editing:** Hyeon Ji Kim, Kyo Hoon Park, Yu Mi Kim, Eunwook Joo, Kwanghee Ahn, Sue Shin.

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
