## [Decision Letter · Decision Letter 0]

2 Oct 2020

PONE-D-20-23433

A protein microarray analysis of amniotic fluid proteins for the prediction of spontaneous preterm delivery in women with preterm premature rupture of membranes at 23 to 30 weeks of gestation

PLOS ONE

Dear Authors,

Thank you for submitting your manuscript to PLOS ONE. After careful consideration, we feel that it has merit but does not fully meet PLOS ONE’s publication criteria as it currently stands. Therefore, we invite you to submit a revised version of the manuscript that addresses the points raised during the review process.

We look forward to receiving your revised manuscript.

Kind regards,

Salvatore Andrea Mastrolia, M.D.

Academic Editor

PLOS ONE

'The funders had no role in study design, data collection and analysis, decision to publish, or preparation of the manuscript.'

Reviewers' comments:

Reviewer's Responses to Questions

**Comments to the Author**

1. Is the manuscript technically sound, and do the data support the conclusions?

Reviewer #1: Yes

2. Has the statistical analysis been performed appropriately and rigorously? 

Reviewer #1: Yes

3. Have the authors made all data underlying the findings in their manuscript fully available?

Reviewer #1: Yes

4. Is the manuscript presented in an intelligible fashion and written in standard English?

Reviewer #1: Yes

5. Review Comments to the Author

Reviewer #1: The subject of this manuscript is novel and definitely can add some new insights to the current knowledge in the field of early diagnosis and prevention of SPTB in pregnancies complicated with PPROM. However, there are some points for consideration in order to improve the quality of this manuscript.

Abstract

1. Objective and methods of abstract are well-written; however, the result part is a little bit confusing and not coherent. It should be giving more details to readers, try re-writing the results while using some statistics.

2. In conclusion part of abstract, you can not give a firm comment (since the subject is novel and further studies are surely needed to confirm the findings). I recommend to use words such as “may or might”. More importantly recommend further studies on this issue.

3. Keywords: the authors have used spontaneous preterm delivery in title and spontaneous preterm birth in keywords alternatively. Also, this alternate use has been done throughout the paper. You should try to be coherent throughout the whole paper with only using one of those terms.

Introduction

4. replace ref 1 with a more recent ref, with updated prevalence of PPROM.

5. Page 4, line 61. please provide ref for definition of early vs late PPROM.

Methods

6. Page 6, line 90. Please define the method you diagnose and confirm PPROM in your population.

7. Page 6, line 102-3. The authors note that the outcome measure was SPTB within 7 and 14 weeks of sampling, how about additionally considering deliveries <34 and <37 and also report data based on these groups? Cases delivering before 34 or 37 and controls after that GA.

8. Page 6, line 107. Please clearly explain the randomization method.

9. Page 10, line 193, ROC was used for sensitivity and specificity, how optimal cut-off was obtained? e.g youden index was used or another formula? Also please remove the “as previously described [22]”, since you have referred to previous studies too much in your methods part, alternatively explain it briefly how cut off was determined.

Results

10. Tables are very informative and clear. Just mark p-values are derived from which test (e.g. student t test or Mann withney) the normality of distribution is an important fact in analyzing the reliability of results.

11. Can analyses also be performed based on GA-defined SPTB e.g. <34 and >34? If your sample size is enough for this analysis, it is highly recommended to do so.

Discussion

12. At the beginning of your discussion, please add 2-3 sentences about the role of inflammation in starting the parturition.

6. PLOS authors have the option to publish the peer review history of their article (what does this mean?). If published, this will include your full peer review and any attached files.

Reviewer #1: **Yes: **Kamran Hessami

---

## [Author Response · Author response to Decision Letter 0]

27 Oct 2020

Manuscript ID: PONE-D-20-23433

October 27, 2020

RE: Manuscript ID: PONE-D-20-23433

“A protein microarray analysis of amniotic fluid proteins for the prediction of spontaneous preterm delivery in women with preterm premature rupture of membranes at 23 to 30 weeks of gestation” 

Dear Editor:

Thank you very much for the review of our manuscript. The comments of the reviewers were constructive and have been used to revise and improve the manuscript.

The following is an itemized account of the changes in the manuscript made in response to comments.

Response to the Editor

Point #1: The Editor made the following comments for us:

Response: According to the editor’s suggestion, we recheck our manuscript and ensure that our paper meets PLOS ONE's style requirements, including those for file naming.

Point #2-a: The Editor suggested that we clarify the sources of funding (financial or material support) for our study and list the grants or organizations that supported our study, including funding received from our institution.

Response: The sources of funding for our study (including the list of the grants or organizations that supported our study) are as follows: 

The current study was supported by the Seoul National University Bundang Hospital Research Fund (Grant No. 13-2020-011), and by the National Research Foundation of Korea (NRF) grant funded by the Korea government (MSIT) (No. 2020R1F1A1048362). The funders had no role in study design, data collection and analysis, decision to publish, or preparation of the manuscript.

We added the above statements to our “Cover letter” as requested.

Point #2-b: The Editor made the following comments for us:

State what role the funders took in the study. If the funders had no role in your study, please state: “The funders had no role in study design, data collection and analysis, decision to publish, or preparation of the manuscript.”

Response: In the current study, the funders had no role in our study. Therefore, we add the following sentence to “Cover letter” as requested. 

Point #2-c: The Editor made the following comments for us:

If any authors received a salary from any of your funders, please state which authors and which funders.

Response: Hyeon Ji Kim, Kyo Hoon Park, Eunwook Joo, and Kwanghee Ahn were employed by Seoul National University Bundang Hospital. Therefore, Hyeon Ji Kim, Kyo Hoon Park, Eunwook Joo, and Kwanghee Ahn have received a salary from Seoul National University Bundang Hospital. We add the aforementioned statement to the “Cover letter”, as requested.

Point #2-d: The Editor made the following comments for us:

If you did not receive any funding for this study, please state: “The authors received no specific funding for this work.” Please include your amended statements within your cover letter; we will change the online submission form on your behalf.

Response: We received funding for this study, as mentioned above. According to the editor’s recommendation, we add our amended statements to the “Cover letter”.

Comments to the Editor and the Reviewer: 

We made a great mistake that the cited supplementary files (Supplementary Materials and Supplementary tables) were not uploaded with the original submission. Thus, we would upload Supplementary files (Supplementary Materials and Supplementary tables) with the revised manuscript files, when submitting revised version. 

Response to the Reviewer #1

Point #1: The reviewer made the following comments for us.

Objective and methods of abstract are well-written; however, the result part is a little bit confusing and not coherent. It should be giving more details to readers, try re-writing the results while using some statistics.

Response: We fully understand the reviewer’s suggestion. Thus, “the result part of the Abstract section” was changed to the following: 

Results: Of all the proteins studied in the protein microarray, four showed significant intergroup differences. Analyses of the total cohort by ELISA confirmed the significantly elevated concentrations of AF lipocalin-2, MMP-9, and S100 A8/A9, but not of endostatin and Fas, in women who delivered within 14 days of sampling. For inflammatory proteins showing a significant association, the odds of SPTD within 14 days increased significantly with an increase in baseline AF levels of the proteins (P for trend <0.05 for each) in each quartile, especially in the 3rd and 4th quartile. 

Point #2: The reviewer made the following comments for us.

In conclusion part of abstract, you cannot give a firm comment (since the subject is novel and further studies are surely needed to confirm the findings). I recommend to use words such as “may or might”. More importantly recommend further studies on this issue.

Response: According to the reviewer’ comments, we add “may” to the last sentence of the conclusion in the Abstract section, and the added sentence is as follows: 

The present findings may highlight the importance of inflammatory mechanisms and the degree of activated inflammatory response in developing SPTD in early PPROM. 

Regarding the reviewer’s comments that further studies are needed on this issue: 

We agree with the reviewer that this point is important. However, the Abstract section should not exceed 300 words in PLOS ONE journal. Thus, we add “further studies on this issue” to the 8th paragraph of the Discussion section and the added sentence is as follows: 

As such, further prospectively designed studies are needed to validate our findings in other cohorts.

The new 8th paragraph of the Discussion section now reads:

The limitations of the current study include its retrospective nature,…………although these observed associations are the same as those in the cohort after excluding the patients analyzed in the discovery phase (Tables S7, S8, S9, and S10). As such, further prospectively designed studies are needed to validate our findings in other cohorts. Another limitation of the study is……………

Point #3: The reviewer made the following comments for us.

Keywords: the authors have used spontaneous preterm delivery in title and spontaneous preterm birth in keywords alternatively. Also, this alternate use has been done throughout the paper. You should try to be coherent throughout the whole paper with only using one of those terms.

Response: It is our fault. We replace “spontaneous preterm birth (SPTB)” with “spontaneous preterm delivery (SPTD)” throughout the whole paper

Point #4: The reviewer recommended replacing “reference 1” with a more recent one, which can provide updated prevalence of PPROM.

Response: As suggested by the reviewer, we add two references, including a reference which provides updated prevalence of PPROM, to the 1st sentence of the Introduction section, and the added references are as follows:

1. Sae-Lin P, Wanitpongpan P. Incidence and risk factors of preterm premature rupture of membranes in singleton pregnancies at Siriraj Hospital. J Obstet Gynaecol Res. 2019;45(3):573-7. 

2. Menon R, Richardson LS. Preterm prelabor rupture of the membranes: A disease of the fetal membranes. Semin Perinatol. 2017;41(7):409-19. 

Point #5: Reviewer recommended adding reference for definition of early vs late PPROM in page 4, line 61. 

Response: As far as we know, in general, PPROM based on the occurrence of gestational age may be classified as pre-viable PPROM (the spontaneous drainage of liquor before 24 weeks of gestation), early PPROM (occurring before 34.0 weeks of gestation), and late PPROM (occurring between 34.0 and 36.6 weeks of gestation).21 Thus, women with PPROM ≥23+0 weeks and ≤30+6 weeks’ gestation in the current study may be considered as having early-PPROM, because gestational age of ≤ 30+6 weeks is also included within the group with gestational age of ≤ 34 weeks. Nevertheless, we believe that it is appropriate to add the definition of “early PPROM” to the Materials and Methods section of the current study, in order to provide the readers clear information about early versus late PPROM (on page 4, line 61). Therefore, we add the following sentence to the 7st paragraph of the Materials and Methods section to define “early PPROM” in the current study.

Based on previous reports [21, 22], women with PPROM ≥ 23+0 weeks and ≤ 30+6 weeks of gestation were defined as having early PPROM.

The new 7st paragraph of the Materials and Methods section now reads:

PPROM was defined as the leakage of AF occurring prior to 37 weeks before the onset of labor ………………… positive nitrazine test. Based on previous reports [21, 22], women with PPROM ≥ 23+0 weeks and ≤ 30+6 weeks of gestation were defined as having early PPROM. Digital examinations were not performed. Management of PPROM,………………

We add the following references to the Reference section with adding the definition of early PPROM.

21. ACOG Practice Bulletin No. 188: Prelabor Rupture of Membranes. Obstet Gynecol. 2018;131(1):e1-e14. Epub 2017/12/22. doi: 10.1097/aog.0000000000002455. PubMed PMID: 29266075.

22. Manuck TA, Rice MM, Bailit JL, Grobman WA, Reddy UM, Wapner RJ, et al. Preterm neonatal morbidity and mortality by gestational age: a contemporary cohort. Am J Obstet Gynecol. 2016 Jul;215(1):103 e1- e14.

Point #6: Reviewer recommended that we define the method we diagnosed and confirmed PPROM in our population on Page 6, line 90. 

Response: In the current study, PPROM was visually diagnosed by examination with a sterile speculum to confirm both pooling of AF in the vagina (or leakage of fluid through the cervix) and a positive nitrazine test. We already described this point (i.e., our methods for PPROM diagnosis) in the 7th paragraph of the Materials and Methods section (on page 9, line 163-166) in our manuscript. Our previous statements regarding the method to diagnose and confirm PPROM in our manuscript (on page 9, line 163-166) are as follows: 

PPROM was defined as the leakage of AF occurring prior to 37 weeks before the onset of labor and was visually diagnosed by examination with a sterile speculum to confirm both pooling of AF in the vagina (or leakage of fluid through the cervix) and a positive nitrazine test.

Point #7: The reviewer made the following comments for us.

On Page 6, line 102-3. The authors note that the outcome measure was SPTB within 7 and 14 weeks of sampling, how about additionally considering deliveries <34 and <37 and also report data based on these groups? Cases delivering before 34 or 37 and controls after that GA.

Response: Good point. As the reviewer’s suggestion, we reanalyze the data based on the occurrence of preterm delivery at <34 weeks of gestation, and the results are shown in the Response to the Point #11. Please refer to the Response to the Point #11. 

Moreover, we add the following statement to the 1st paragraph of the Materials and Methods section with adding an additional analysis of preterm delivery <34+0 weeks as an outcome measure.

An additional analysis was performed for preterm delivery at <34+0 weeks. 

On the other hand, we cannot consider preterm delivery at < 37 weeks as an outcome measure because based on the practices for PPROM at our hospital, labor was induced after 34 weeks of gestation or greater (but before 370/7 weeks of gestation) in most patients with PPROM, as described in the 7th paragraph of the Materials and Methods section (management of PPROM). Thus, we do not reanalyze the data based on the occurrence of preterm delivery at <37 weeks of gestation. 

Point #8: Reviewer suggested that we explain the randomization method. 

Response: For the discovery phase using the antibody microarray technique, we randomly selected 15 cases from the list of 45 women with SPTD within 14 days of sampling in a cohort of 87 women with PPROM. Following are details on the methods employed for random selection of cases: we (1) created serial numbers for the list of 45 women with SPTD within 14 days, which was prepared in a Microsoft Excel sheet; (2) generated a random sequence of numbers between 1 and 45, using the random sequence generator application; and (3) selected 15 cases, according to the first 15 numbers generated using the random sequence generator. Thereafter, each control-woman was chosen for each case-woman, matched by gestational age at sampling, maternal age, parity, and length of specimen storage. Thus, the previous description about the selection of control-women was not accurate in the manuscript because control-women were not randomly chosen, but matched to the case-women. To clearly explain the randomization method, therefore, the 2nd sentence of the 2nd paragraph in the Materials and Methods section is changed as follows:

These case-women were randomly chosen from the list of 45 women who delivered within 14 days of sampling in a cohort of 87 women with PPROM, using a random sequence generator, whose details are described in the Supplementary Materials. Each control-woman was chosen for each case-woman, matched by gestational age at sampling, maternal age, parity, and length of specimen storage. 

We add “the following methods employed for random selection of cases” to the Supplementary Materials section.

Methods employed for random selection of cases

Following are details on the methods employed for random selection of cases: we (1) created serial numbers for the list of 45 women with SPTD within 14 days, which was prepared in a Microsoft Excel sheet; (2) generated a random sequence of numbers between 1 and 45, using the random sequence generator application; and (3) selected 15 cases, according to the first 15 numbers generated using the random sequence generator.

Point #9: The reviewer made the following comments for us.

Page 10, line 193, ROC was used for sensitivity and specificity, how optimal cut-off was obtained? e.g youden index was used or another formula? Also please remove the “as previously described [22]”, since you have referred to previous studies too much in your methods part, alternatively explain it briefly how cut off was determined.

Response: In the current study, the highest Youden index that gave the maximum sum of sensitivity and specificity was used to determine best cut-offs in ROC curves. Thus, as the reviewer suggested, the 3rd sentence in the last paragraph of the Materials and Methods section (on page 10, line 191) is changed as follows: 

Receiver operating characteristic (ROC) curves for predicting SPTD were used to identify optimal cut-off values based on the maximum Youden index [maximum (sensitivity + specificity - 1)] for each studied protein in the AF, to assess the diagnostic accuracy of each protein, and to compare the usefulness of different proteins in the same patients using the method proposed by DeLong et al [22]

In addition, we replace “as previously described [22]” with “using the method proposed by DeLong et al [22]” in the 3rd sentence of the last paragraph of the Materials and Methods section.

Point #10: Reviewer suggested that we just mark p-values derived from Student-t test or Mann-Whitney U test because the normality of distribution is an important fact in analyzing the reliability of results.

Response: Good point. According to the reviewer’s suggestion, we add “the mark of p-values (e.g., derived from Student-t test or Mann-Whitney U test)” to the Table 1, 3, 4, S1, S2, S4, S5, S7, and S8 as the superscript a, b, and c, and provided the explanation in the footnotes of these Tables. 

Point #11: Reviewer asked us to re-analyze the data based on the SPTB at <34 versus ≥34 weeks of gestation. 

Response: Excellent point. According to the reviewer’s suggestion, we reanalyze the data based on the occurrence of preterm delivery at <34+0 weeks of gestation, and the table with univariate test result is as follows: 

Table S5 Characteristics of the study population grouped by preterm delivery at < 34 weeks in the total cohort 

Variables Preterm delivery at < 34+0 weeks

(n = 71) Preterm delivery at ≥ 34+0 weeks (n=17) 

Maternal age (years) 32.1 ± 3.6 31.7 ±3.9 0.716a

Nulliparity 39..4% (28/71) 58.8% (10/17) 0.147c

Gestational age at sampling (weeks) 27.6 ± 2.3 27.8 ± 2.2 0.824b

Gestational age at delivery (weeks) 29.5 ± 2.2 35.5 ± 2.2 < 0.001b

Sampling-to-delivery interval (days) 13.3 ± 13.9 53.5 ± 22.5 <0.001b

AF endostatin (ng/mL) 68.7 ± 26.0 63.9 ± 18.7 0.788b

AF Fas ( ng/mL) 5.02 ± 1.94 4.40 ± 1.69 0.163b

AF IL-8 (ng/mL) 7.2 ± 6.4 2.1 ± 4.1 <0.001b

AF lipocalin-2 (µg/mL) 1.27 ± 0.96 0.59 ± 0.79 0.004b

AF MMP-9 (ng/mL) 79.74 ± 91.03 25.11 ± 57.01 <0.001b

AF S100 A8/A9 (µg/mL) 21.6 ± 22.7 11.4 ± 18.9 0.023b

Positive AF cultures 54.9% (39/71) 5.9% (1/17) <0.001c

Use of tocolytic agents 76.1% (54/71) 47.1% (8/17) 0.019c

Use of antibiotics 97.2% (69/71) 88.2% (15/17) 0.167c

Use of antenatal corticosteroids 97.2% (69/71) 58.8% (10/17) <0.001c

Clinical chorioamnionitis 16.9% (12/71) 5.9% (1/17) 0.448c

Histological chorioamnionitis d 73.2% (52/71) 28.6% (4/14) 0.004c

AF, amniotic fluid; Fas (TNFRSF6), ﬁbroblast-associated (tumor necrosis factor receptor superfamily member 6); IL, interleukin; MMP, matrix metalloproteinase; S100 A8/A9, S100 calcium binding protein A8/A9 complex. 

Data are given as the mean ± standard deviation or % (n/N). 

a Student’s t-tests

b Mann-Whitney U-tests 

c χ2-tests or Fisher’s exact tests, where appropriate. 

d Three cases were excluded for the analysis because delivery took place at another institution.

We add the Table above to the Supplementary table section as a new Table S5.

As shown in the above Table, the median levels of lipocalin-2, S100 A8/A9, IL-8, and MMP-9 were significantly higher in AF from women with preterm delivery at < 34 weeks than in those who delivered at ≥ 34 weeks. Moreover, the rates of use of tocolytic agents and corticosteroids were significantly higher in women with preterm delivery at < 34 weeks. Therefore, multivariate logistic regression analyses were further performed to examine the independent association between the AF levels of 4 target proteins and preterm delivery at < 34 weeks, after adjusting for baseline clinical parameters (i.e., use of tocolytic agents and corticosteroids), with a P value < 0.05 from the univariate analysis, as previously described in the Statistical section. However, significant association between 4 target proteins in AF and preterm delivery at < 34 weeks, as shown in univariate analyses, disappeared upon adjusting for baseline variables (i.e., use of tocolytics and corticosteroids). The Table regarding Results of multivariable regression analyses for predicting preterm delivery at < 34 weeks is below. 

Table S6. Multivariable logistic regression model showing the unadjusted and adjusted odds ratios of association between potential amniotic fluid proteins and preterm delivery at < 34 weeks in women with preterm premature rupture of membranes in the total cohort (n = 88)

Variables Odds ratio (95% confidence interval) 

 Unadjusted Adjusteda P-valueb

AF IL-8 (ng/mL) 1.255 (1.048 - 1.432) 1.161 (0.990 – 1.361) 0.066

AF lipocalin-2 (µg/mL) 2.567 (1.189 – 5.544) 2.285 (0.959 – 5.448) 0.062

AF MMP-9 (ng/mL) 1.010 (1.001 – 1.019) 1.009 (0.999 – 1.020) 0.078

AF S100 A8/A9 (µg/mL) 1.027 (0.995 – 1.060) 1.016 (0.982 – 1.052) 0.354

AF, amniotic fluid; IL, interleukin; MMP, matrix metalloproteinase; S100 A8/A9, S100 calcium binding protein A8/A9 complex.

a For use of tocolytics and corticosteroids. 

b Of odds ratio adjusted for use of tocolytics and corticosteroids.

We add the Table above (multivariable logistic regression model) to the Supplementary tables section as a new Table S6.

We add the following paragraph at the end of the Results section with adding the results using preterm delivery at < 34 weeks as endpoint. 

AF biomarkers with respect to preterm delivery at < 34+0 weeks 

When preterm delivery at < 34 weeks of gestation was used as the outcome measure, univariate analyses yielded the same results for the measured AF proteins as those obtained using SPTD within 14 days of sampling (Supplemental S5 Table). However, the rates of use of tocolytics and corticosteroids were significantly higher in women who subsequently had preterm delivery at <34 weeks than in women who delivered at ≥34 weeks. Thus, multivariate analyses were performed, which revealed that significant associations between the four proteins in AF and preterm delivery at < 34 weeks, which was corroborated by univariate analyses, disappeared upon adjusting for baseline variables (i.e., use of tocolytics and corticosteroids) in multivariate analyses (Supplemental Table S6).

We add the two Tables above to the Supplementary Tables section as new Tables S5 and S6. With adding the new Tables S5 and S6, we change “the numbers of the original Tables (S5, S6, S7, and S8)” to “the new numbers of corresponding Tables (S7, S8, S9, and S10)” in the Supplementary Tables section and throughout Manuscript. 

Point #12: Reviewer asked us to add 2-3 sentences about the role of inflammation in starting the parturition at the beginning of Discussion section. 

Response: According to the reviewer’s suggestion, we add the following sentences to the 2nd paragraph of the Discussion section to describe the role of inflammation in starting the parturition.

Evidence suggests that inflammation is an essential component in the initiation of term as well as preterm parturition [26]. Specifically, elevated IL-6, IL-8, and MMP-8 levels in the amniotic cavity have been demonstrated to be associated with the onset of preterm and term labor, along with membrane rupture [16, 26-29]. Furthermore, the present study has shown that the severities of SPTD are associated with the gradations in inflammatory response in the AF based on the levels of multiple inflammatory markers. 

We add the following references to the Reference section with adding the statements that describe “the role of inflammation in starting the parturition”.

16. Lee SY, Park KH, Jeong EH, Oh KJ, Ryu A, Kim A. Intra-amniotic infection/inflammation as a risk factor for subsequent ruptured membranes after clinically indicated amniocentesis in preterm labor. Journal of Korean medical science. 2013;28(8):1226-32. Epub 2013/08/21. doi: 10.3346/jkms.2013.28.8.1226. PubMed PMID: 23960452; PubMed Central PMCID: PMCPMC3744713.

26. Romero R, Espinoza J, Goncalves LF, Kusanovic JP, Friel LA, Nien JK. Inflammation in preterm and term labour and delivery. Semin Fetal Neonatal Med. 2006;11(5):317-26. Epub 2006/07/15. doi: 10.1016/j.siny.2006.05.001. PubMed PMID: 16839830.

27. Saito S, Kasahara T, Kato Y, Ishihara Y, Ichijo M. Elevation of amniotic fluid interleukin 6 (IL-6), IL-8 and granulocyte colony stimulating factor (G-CSF) in term and preterm parturition. Cytokine. 1993;5(1):81-8. Epub 1993/01/01. doi: 10.1016/1043-4666(93)90027-3. PubMed PMID: 7683506.

28. Biggio JR, Jr., Ramsey PS, Cliver SP, Lyon MD, Goldenberg RL, Wenstrom KD. Midtrimester amniotic fluid matrix metalloproteinase-8 (MMP-8) levels above the 90th percentile are a marker for subsequent preterm premature rupture of membranes. Am J Obstet Gynecol. 2005;192(1):109-13. Epub 2005/01/27. doi: 10.1016/j.ajog.2004.06.103. PubMed PMID: 15672011; PubMed Central PMCID: PMCPMC3721523.

29. Maymon E, Romero R, Pacora P, Gomez R, Athayde N, Edwin S, et al. Human neutrophil collagenase (matrix metalloproteinase 8) in parturition, premature rupture of the membranes, and intrauterine infection. Am J Obstet Gynecol. 2000;183(1):94-9. Epub 2000/08/02. doi: 10.1067/mob.2000.105344. PubMed PMID: 10920315.

I hope that our revised manuscript and additional information that we provided in this letter meets with your approval. Please let us know if you have further questions or require additional information. I appreciate your time and assistance. 

With regards,

Kyo Hoon Park, MD, PhD

Professor

Department of Obstetrics and Gynecology. Seoul National University Bundang Hospital

166 Gumiro, Seongnamsi, Kyeonggido, 463-707, Korea

Tel: 82-31-787-7252

Fax: 82-31-787-4054

E-mail: pkh0419@snubh.org

Enclosures

---

## [Decision Letter · Decision Letter 1]

16 Dec 2020

A protein microarray analysis of amniotic fluid proteins for the prediction of spontaneous preterm delivery in women with preterm premature rupture of membranes at 23 to 30 weeks of gestation

PONE-D-20-23433R1

Dear Dr. Authors,

We’re pleased to inform you that your manuscript has been judged scientifically suitable for publication and will be formally accepted for publication once it meets all outstanding technical requirements.

Kind regards,

Salvatore Andrea Mastrolia, M.D.

Academic Editor

PLOS ONE

Additional Editor Comments (optional):

Reviewers' comments:

Reviewer's Responses to Questions

**Comments to the Author**

1. If the authors have adequately addressed your comments raised in a previous round of review and you feel that this manuscript is now acceptable for publication, you may indicate that here to bypass the “Comments to the Author” section, enter your conflict of interest statement in the “Confidential to Editor” section, and submit your "Accept" recommendation.

Reviewer #1: All comments have been addressed

2. Is the manuscript technically sound, and do the data support the conclusions?

Reviewer #1: Yes

3. Has the statistical analysis been performed appropriately and rigorously? 

Reviewer #1: Yes

4. Have the authors made all data underlying the findings in their manuscript fully available?

Reviewer #1: Yes

5. Is the manuscript presented in an intelligible fashion and written in standard English?

Reviewer #1: Yes

6. Review Comments to the Author

Reviewer #1: Authors have responded to all comment. no further comments. Hope authors of this study continue the path on this subject .

7. PLOS authors have the option to publish the peer review history of their article (what does this mean?). If published, this will include your full peer review and any attached files.

Reviewer #1: **Yes: **Kamran Hessami

---

## [Editor Report · Acceptance letter]

21 Dec 2020

PONE-D-20-23433R1 

A protein microarray analysis of amniotic fluid proteins for the prediction of spontaneous preterm delivery in women with preterm premature rupture of membranes at 23 to 30 weeks of gestation 

Dear Dr. Park:

I'm pleased to inform you that your manuscript has been deemed suitable for publication in PLOS ONE. Congratulations! Your manuscript is now with our production department. 

Kind regards, 

on behalf of

Dr. Salvatore Andrea Mastrolia 

Academic Editor

PLOS ONE